# Electrochemical Deposition of ZnO Nanowires on CVD-Graphene/Copper Substrates

**DOI:** 10.3390/nano12162858

**Published:** 2022-08-19

**Authors:** Issam Boukhoubza, Elena Matei, Anouar Jorio, Monica Enculescu, Ionut Enculescu

**Affiliations:** 1Group of Nanomaterials and Renewable Energies, Laboratory of Solid State Physics, Faculty of Sciences Dhar El Mahraz, Sidi Mohammed Ben Abdellah University, P.O. Box 1796, Atlas Fez 30000, Morocco; 2National Institute of Materials Physics, Atomistilor 405A, 077125 Magurele, Romania

**Keywords:** CVD-graphene, ZnO nanowires, electrochemical deposition, photoluminescence, optoelectronic properties

## Abstract

ZnO nanostructures were electrochemically synthesized on Cu and on chemical vapor deposited (CVD)-graphene/Cu electrodes. The deposition was performed at different electrode potentials ranging from −0.8 to −1.2 V, employing a zinc nitrate bath, and using voltametric and chronoamperometric techniques. The effects of the electrode nature and of the working electrode potential on the structural, morphological, and optical properties of the ZnO structures were investigated. It was found that all the samples crystallize in hexagonal wurtzite structure with a preferential orientation along the c-axis. Scanning electron microscopy (SEM) images confirm that the presence of a graphene covered electrode led to the formation of ZnO nanowires with a smaller diameter compared with the deposition directly on copper surface. The photoluminescence (PL) measurements revealed that the ZnO nanowires grown on graphene/Cu exhibit stronger emission compared to the nanowires grown on Cu. The obtained results add another possibility of tailoring the properties of such nanostructured films according to the specific functionality required.

## 1. Introduction

Quasi-one-dimensional ZnO nanostructures, such as nanowires, nanorods, and nanotubes are seen to be interesting subjects of research since these can be used as building blocks for a wide range of applications [1,2,3,4]. ZnO is a semiconducting material (usually n type) with a set of properties which include high transparency, wide direct bandgap (3.37 eV), large exciton binding energy of 60 meV, and an electron mobility which can reach 155 cm^2^ V^−1^s^−1^ [5,6,7]. The material is rather abundant, non-toxic, and radiation resistant, these qualities making it an interesting material for numerous electronic and optoelectronic applications such as sensors [8], transistors [1,9], supercapacitors [10,11], photovoltaic devices [12], and photocatalytic structures [13]. A large number of reports dealing with the synthesis of ZnO nanostructures is available, simple, low-cost preparation methods such as hydrothermal method [14], spin-coating [15], electrochemical deposition [2,16], and chemical deposition [17,18] being preferred. It was found that for this particular oxide the synthesis parameters are directly influencing the micromorphology, ZnO being a polymorphic material. Platelets, rods, tubes, wires, prisms, and tubes are obtained.

Low dimensional carbon structures have opened new opportunities for applications due to their specific properties. In particular, graphene, a monolayer of carbon atoms with a 2D hexagonal crystal lattice has a wide range of excellent properties, such as high electron mobility [19], optical transparency [20], and increased mechanical and chemical stability [21] and is therefore suitable for electronic or optoelectronic applications [22,23,24]. Different techniques have been employed for graphene fabrication including processes such as chemical vapor deposition (CVD) [25], annealing SiC substrates [26], liquid-phase exfoliation (LPE), and mechanical exfoliation [27,28], each process producing different amounts of material with different quality parameters. Among these methods, the growth by CVD employs metallic substrates as catalysts and a hydrocarbon gas as carbon source (CH_4_). The process is based on carbon surface adsorption on the metal substrate with low carbon solubility (Cu). CVD growth of graphene results in single or few layers of material deposition, usually into a 2-dimensional polycrystalline film. The graphene CVD process on metal substrates has shown good results in producing single-layer graphene on a large surface with high quality in terms of structure and purity [29,30].

In recent years, ZnO nanostructures/CVD-graphene hybrids and nanocomposites have been synthesized to improve the performance of photocatalyst [31] and optoelectronic applications such as supercapacitor, sensor, light-emitting diodes, and photodetectors [11,32,33,34,35,36]. In particular, Brent et al. [37] used a hydrothermal floating technique to deposit ZnO nanorods on CVD-graphene to investigate the effects of the microstructure and interface on the performance of ZnO nanorods/CVD-graphene UV photodetectors. Ju et al. [38] constructed membrane-based flexible photodetectors for imaging applications, based on a structure of Ag nanowires/ZnO nanowires/graphene in which ZnO nanowires were grown by CVD, mixed with Ag nanowires, and used as a film. Ge et al. [39] reported the fabrication of ZnO nanorods on CVD-graphene surface by hydrothermal method for simultaneous electrochemical sensing. Hak et al. [40] demonstrated the synthesis of vertically aligned ZnO NWs on CVD-graphene/Cu by using self-catalyzed vapor-phase transport technique and developed graphene/ZnO NR/graphene heterojunction devices. Usually, the fabrication of this type of composites takes place in two steps: a first step for the growth of graphene on metal substrate and a second step for the deposition of ZnO nanostructures on graphene surface. Various techniques were employed for depositing ZnO on graphene such as hydrothermal technique and self-catalyzed vapor-phase transport. There are few published papers on the growth of ZnO nanowires on graphene-covered substrates by employing electrochemical deposition [41,42,43]. Electrodeposition is an interesting technique for preparing thin semiconductor films since it is not based on expensive equipment and provides good control over parameters such as thickness, morphology, crystallite size, and aspect ratio of the deposit. By tailoring the process parameters, such as bath composition, temperature, deposition time, potential, or current, deposits can be obtained with a wide range of morphologic or structural characteristics [2,44,45]. In this way, nanostructures which include nanowires, nanorods, prisms or platelets can be fabricated in a controlled manner. Using CVD-graphene on Cu substrates can improve the control on the morphology, density, and orientation of the ZnO nanostructures in an electrodeposition process.

In this work, an approach in fabricating tailored ZnO nanowires on Cu and CVD-graphene/Cu films working electrodes is described. A CVD process was employed for fabricating graphene on copper substrates. Different electrode potentials were used during the electrochemical deposition process. A wide range of techniques, including scanning electron microscopy (SEM), X-ray diffraction (XRD), Raman spectroscopy, as well as reflectance (R%) and photoluminescence (PL) spectroscopy, were used to study the effect of the electrode potential and graphene surface on the structural, morphological, and luminescence properties of ZnO/Cu and ZnO/G/Cu films.

## 2. Materials and Methods

### 2.1. Synthesis of Graphene, ZnO/Cu, and ZnO/G/Cu Films

Graphene (G) was prepared on polycrystalline copper (Cu) substrates (99.99+% purity from Goodfellow Cambridge Limited, Huntingdon, England) of 25 µm thickness, by employing the chemical vapor deposition method (CVD). An As-One Rapid Thermal Processor system produced by Annealsys (Montpellier, France) was employed. The Cu substrates (0.5 cm × 2 cm) were cleaned by ultrasonication in water with detergent, deionized water, acetone, and isopropanol respectively. A typical procedure was employed, starting by placing the Cu substrates in the deposition chamber and annealing it at 900 °C in a H_2_ (10 sccm) atmosphere for 15 min at a pressure of 10 mbar. Further, the graphene was grown on the metal surface using a gas reaction mixture of H_2_:CH_4_ with a flow ratio of 10:25 being injected in as carbon (C) source. This process was performed for 15 min at a temperature of 900 °C and a pressure of 10 mbar and was followed by one of rapid cooling in Ar atmosphere. The samples were then removed from the deposition chamber and used as working electrodes as prepared.

ZnO electrodeposition was performed using an aqueous solution with a composition of Zn(NO_3_)_2_ · 6H_2_O and 100 mM KNO_3_. The bath temperature was kept constant using a thermostat and a double wall electrochemical cell during the process. For ZnO nanostructures electrodeposition, a three-electrodes system was used, consisting of a Pt mesh woven as a counter electrode (CE), a potassium chloride saturated Red-Rod electrode as reference (RE), and Cu or G/Cu (0.5 cm × 2 cm) as working electrodes (WE). The electrochemical deposition processes were performed using a Voltalab 10 PGZ 100 potentiostat/galvanostat controlled by computer with Volta Master 4 Electrochemical Software from Radiometer Analytical SAS (Lyon, France). ZnO nanowires growth on Cu and CVD-G/Cu was carried out at different potential values versus the RE at a temperature of 90 °C, these conditions being found to be optimal in order to achieve the highest density of vertically aligned wires. The electrochemical polarization measurements were performed in order to understand the electrodeposition process of ZnO on Cu and G/Cu. The ZnO NWs/Cu and ZnO NWs/G/Cu samples were obtained using potentiostatic deposition, with applied potentials: −0.8 V, −0.9 V, −1 V, −1.1 V, −1.2 V vs. RE without stirring. After the electrodeposition, the as-deposited films were rinsed with distilled water and dried. The preparation procedure is shown in Figure 1.

### 2.2. Characterization

The structural characterization of the films was performed by X-ray diffraction measurements performed with a Bruker D8 Advance X-ray diffractometer (Bruker AXS GmbH, Karlsruhe, Germany) using CuKα radiation (λ = 1.54178 Å) in the 20–70° range. The surface morphology of ZnO nanostructures/Cu and ZnO/Graphene/Cu was analyzed using a Gemini 500 field emission scanning electron microscope from Zeiss (Oberkochen, Germany). The Raman spectra of the ZnO/Cu and ZnO/G/Cu films were recorded at room temperature using a LabRAM HR Evolution Raman spectrometer (Horiba Jobin-Yvon: Edison, NJ, USA), with a He–Ne laser functioning at 633 nm being focused by an Olympus 100× objective on the surface of the films. The reflectance spectra were recorded using a PerkinElmer Lambda 45 UV–Vis spectrophotometer(Waltham, Massachusetts, USA) and the photoluminescence measurements were performed using an Edinburgh FL 920 PL spectrometer (Edinburgh Instruments, Livingston, UK), with 325 nm UV excitation at room temperature.

## 3. Results and Discussion

### 3.1. Electrochemical Study

The electrochemical polarization curves recorded for ZnO deposition on Cu and G/Cu presented in Figure 2a show both similarities and differences between the two types of working electrodes. It is known that the electrochemical deposition of ZnO is a two-step process with a first step being the nitrate to nitrite reduction resulting in local OH^−^ ions production, the second step being the precipitation of Zn(OH)_2_ which further decomposes into ZnO [46,47]. The electrochemical current measurements are therefore not a direct measurement of the deposition process but an indirect one through the rate of nitrate reduction. The polarization curves estimate the evolution of the reduction potential on Cu and graphene/Cu working electrodes, respectively. Figure 2a illustrates the differences between the Cu and graphene/Cu working electrodes in ZnO nanowires electrodeposition. Two observations can be made, the deposition on copper electrode generates a higher current and a wide peak at about −550 mV. This may be related to a pronounced catalytic effect of the copper substrates on the reduction reaction taking place. Furthermore, the conclusion is supported by the fact that the main deposition range starts at a more anodic potential for copper working electrodes, the difference being of about 100 mV.

For the electrodeposition of ZnO nanowires on Cu and graphene/Cu using chronoamperometry, several different potentials from −0.8 V to −1.2 V were selected using a deposition bath containing 5 mM of Zn(NO_3_)_2_·6H_2_O and 100 mM KNO_3_. Chronoamperometric curves presented in Figure 2b,c for the two working electrodes and for the full range of deposition potentials show a behavior of a nucleation–growth mechanism, a tendency which is more pronounced for the graphene-covered electrode. As was observed already from the electrochemical polarization curves, the current density is higher for the case of simple copper electrodes—i.e., a higher nitrate reduction rate at similar electrode potential. A current decrease can be observed after an initial sharp peak in the first seconds due to the formation of the double layer and due to the nucleation process [48]. For the graphene-covered electrode the process is slower, a slight decrease in current being observed throughout the experiment.

The near constant current region corresponds to a stable deposition regime, the deposition rate depending on bath composition and on the electrode potential. When increasing the applied potential, an increase in current density is observed.

### 3.2. Structural Properties

The X-ray diffraction patterns of ZnO/Cu and ZnO/G/Cu films deposited at applied potentials of −0.8 V, −0.9 V, −1 V, and −1.1 V are shown in Figure 3a,b. The characteristic peaks at 31.89°, 34.56°, 36.4°, 47.57°, 56.6°, and 62.9°, correspond to the (100), (002), (101), (102), (110), and (103), crystal planes of hexagonal wurtzite structure of ZnO (JCPDS NO.36-1451). These planes were observed in all samples with different relative intensities.

In all cases, two peaks at 43° and 51° were observed, which correspond to the (111) and (200) planes of the Cu substrate [29]. Furthermore, the intensities of (002) peaks for all samples were the strongest (Figure 4a), different from the powder standard, which indicates that the ZnO nanowires have a preferred orientation along the c-axis direction, usual for this particular morphology and for these experimental parameters. However, it is easy to notice that the ratio between the (002) peak and the other peaks is higher for the graphene-covered electrodes. For the samples obtained at −0.9 V deposition potential on graphene covered electrode, other peaks are almost invisible showing an almost perfect orientation of the nanowires. Corroborating this observation to the electrochemical data, one can conclude that on graphene, the nucleation process is slower but also the nuclei are preferentially oriented along the (002) direction.

Based on the full-width-at-half-maximum (FWHM) data for the (101), (002), and (100) diffraction peaks (Table 1), the average crystallite size of the ZnO nanowires was calculated using the Debye–Scherrer formula, as given in Equation (1) [49]:(1)$$D\;=\frac{k\;\lambda }{\beta \cos\theta }\;\;$$
where, k = 0.89 is Scherer’s factor, λ is the X-ray wavelength (λ = 1.54178 Å, CuKα radiation), θ is the Bragg angle, and β the line broadening at half height of the maximum intensity (FWHM), measured in radians. The mean crystallite size of the ZnO nanowires corresponding to the FWHM of the most intense peak at 34.56° showed that ZnO/G/Cu films (66.19 nm, 58.35 nm, 48.57 nm, 49.76 nm) was smaller than synthesized on Cu (68.9 nm, 65.65 nm, 56.14 nm, 57.13 nm) (Figure 4b). Additionally, the FWHM value of the highest intense peak (002) of ZnO grown on G/Cu was a little smaller than ZnO/Cu, which suggests that the layer of graphene impacted the growth of ZnO nanowires most probably by decreasing the diameter. It is also important to point out that, as the deposition potential is more electronegative, the crystallite size decreases as a consequence of the induced change in deposition mechanisms from a reaction rate limitation to a diffusion rate limitation.

The preferential orientation was estimated from the XRD measurements by calculating the texture coefficient for a (*hkl*) plan (TC (*hkl*)) by using the Equation (2) [50]:(2)$$\mathrm{TC}\left(hkl\right)=\frac{I\left(hkl\right)/I_{0}\left(hkl\right)}{\frac{1}{N}\sum_{1}^{N}I\left(hkl\right)/I_{0}\left(hkl\right)}$$
*I*(*hkl*) is the measured intensity, *I*_0_(*hkl*) is the standard intensity from the (JCPDS NO.36-1451) and n is the number of reflections; three diffraction (N = 3) planes being chosen (100), (002), (101). Figure 5a,b shows the texture coefficient (TC) of (100), (002), and (101) planes for ZnO/Cu and ZnO/G/Cu at a different applied potential. The high TC of (002) values in all samples confirms that the ZnO nanowires exhibit a preferential orientation growth along the c-axis. Moreover, the (002) texture is higher in the graphene covered electrode samples than in the samples grown on simple copper electrodes.

### 3.3. Morphology Analysis

Figure 6 and Figure 7 show SEM images of ZnO nanowires grown by electrochemical deposition on Cu and CVD–graphene/Cu electrodes at different electrode potential values. According to SEM images of ZnO nanowires/Cu (Figure 6a–e), it can be observed that ZnO nanowires are dense and homogeneous, completely coating the Cu electrode. In both cases, one can notice a well-crystalized material. In addition, the SEM images of ZnO/Cu at −0.8 V and −0.9 V (Figure 6a,b) show the presence of nanowires with hexagonal and planar face at their top. The analysis of SEM images at −1 V, −1.1 V, and −1.2 V reveals the presence of nanowires with a pyramid face at their top. This is the result of a change induced by the difference between a reaction rate-limited process at less electronegative potential and a diffusion rate-limited process at more electronegative potential.

In the case of CVD-graphene/Cu electrodes, (Figure 7a) the ZnO nanowires obtained follow the same tendencies in terms of diameter/crystallite size. It is interesting to note that when the deposition takes place in the range of potentials of −0.8 V to −1.1 V (Figure 7a–d), a high density of wires can be observed, this decreasing in the case of an electrode potential of −1.2 V (Figure 7e). This behavior is again consistent with a nucleation growth model and a diffusion rate limited process. The cross-section SEM images (e.g., Figure 7f) confirm the formation of ZnO nanowires/graphene/Cu film at −0.9 V. In this particular case, the length of the ZnO nanowires is 1.5–2 μm.

The average diameter distribution was determined by measuring ZnO nanowires from SEM images using Image J software (Figure 8). This figure shows nanowire diameter with a size distribution centered between 20 and 300 nm. Moreover, the average size in diameter of ZnO nanowires grown on G/Cu is lower compared to those of ZnO grown on Cu substrates (Figure 8k). Furthermore, as the potential becomes more electronegative the average diameter becomes smaller in both cases, copper and graphene covered electrodes. In addition, according to the obtained XRD results, the nanowires grown on the graphene surface are more vertically aligned than ZnO/Cu (see Figure 5). As a result, the graphene surface plays an important role in the growth, orientation, density, and diameters of electrodeposited ZnO nanostructures.

### 3.4. Raman Study

The uniformity and structural quality of the electrodeposited nanostructures were also analyzed from the Raman spectroscopy measurements at room temperature. Figure 9 shows the Raman spectra of the ZnO/G/Cu films deposited at different potentials. For all Raman spectra of ZnO/G/Cu, three peaks are observed at 436 cm^−1^, 1308 cm^−1^, and 1600 cm^−1^. The first peak at 436 cm^−1^ ($E_{2}^{High}$ mode) is associated to oxygen vibration of ZnO [50]. The second peak at 1308 cm^−1^ corresponds to the D band of graphene attributed to layer disorder due to the level of defects. The third peak at 1600 cm^−1^ corresponds to the graphene G band originating from in-plane stretching vibration of sp^2^ carbon and is a double degenerate phonon mode (*E*_2g_ symmetry) [51,52]. The spectra of ZnO/G/Cu deposited at −1.1 V, includes a new peak at 2656 cm^−1^ assigned to 2D band of graphene. According to the literature, the 2D band is related to the number of graphene sheets being visible only in this case due to the lowest nanowire density.

Comparing the Raman spectra between the samples grown on the two types of electrodes (Figure 10a,b), the growth of ZnO on graphene shifts the position of the peak $E_{2}^{High}$ to higher frequencies (from 435 to 438 cm^−1^) and modifies its intensity.

Furthermore, the full width at half maximum (FWHM) of the peak at 438 cm^−1^ ($E_{2}^{High}$ mode) decreases in the ZnO/Cu when compared to ZnO/G/Cu samples most probably due to the covalent interaction of the ZnO with the graphene layers (Figure 10a,b). According to data analysis from literature, Biroju et al. [53] noted that lower $E_{2}^{high}$ in ZnO wires, as is the case for the ZnO deposited on graphene covered electrodes, indicate a better crystalline structure.

The crystallite size of graphene used for ZnO electrodeposition can be obtained from Raman spectroscopy measurements. Thus, the I_D_/I_G_ ratio obtained from the Raman spectra on ZnO/G/Cu samples was used to calculate the average crystallite size (*L_a_*) of graphene layer (Table 2), using the Equation (3) [54]:(3)$$L_{a}\left(\mathrm{nm}\right)=\left(2.4\times 10^{-10}\right)\lambda^{4}{\left(\frac{I_{D}}{I_{G}}\right)}^{-1},$$
where λ is the laser excitation (633 nm).

### 3.5. Optical Properties

The UV–Vis total reflectance spectra of ZnO/Cu and ZnO/G/Cu are shown in Figure 11, a blue shift being observed in the band for the graphene-covered electrodes.

The inset of Figure 11 shows the estimation of the band gap of samples by calculating the slopes of reflectance (R) spectra based on Kubelka–Munk function F(R), using Equation (4):(4)$$F\left(R\right)=\frac{{\left(1-R\right)}^{2}}{2R}$$

The band gap energy of ZnO/Cu was found to be 3.15 eV, which is slightly less than the band gap of ZnO/G/Cu (3.21 eV) (inset Figure 11). This can be induced by the interaction between ZnO nanowires and graphene layer and to the concentration of point defects in the material. Nanakkalt et al. [55] reported that such a blue shift in the absorption band and in the energy band gap can be associated to the differences in concentration of point defects induced by the presence of graphene. In any case, these two films have a significantly lower band gap than previously reported for ZnO (3.37 eV) [7].

### 3.6. Photoluminescence

In Figure 12, the photoluminescence (PL) spectra of ZnO NWs/Cu and ZnO/G/Cu obtained using a 325 nm excitation are presented. The typical ZnO PL spectra exhibit two distinct types of bands [44,56]. The near band to band emission (NBE) is located around ~376–378 nm (UV emission) and is attributed to free exciton recombination while the other bands are related to various point defects. In the present case, the broad visible emission band in the range of 580–602 nm can be assigned to oxygen-related defects. The PL spectra plotted show that in all the cases, the wires grown on graphene present a more intense excitonic peak when compared to ZnO nanostructures grown directly on copper at the same electrode potential. The observation is related to a lower concentration of defects and can be correlated to the results obtained from the diffraction measurements. Defect concentration is also related to the deposition potential, the luminescence of the exciton peak being more intense for the samples grown at less electronegative potential. The results are consistent with the fact that for a lower growth rate, there is a lower concentration of defects in the material. For the case of the samples grown at the most electronegative potential, the excitonic peak almost disappears, being dwarfed by the defect-related bands. As expected, the only case where the excitonic peak is more intense than the defect-related band is at the lowest deposition rate, i.e., at the least electronegative potential (−0.8 V). On the other hand, the PL intensity ratio of the UV emission to the visible emission (I_uv_/I_vis_) (Tabel 3), meaning the ratio between the intensity of the exciton luminescence and the defect related luminescence, was found for all samples below one except for the structures grown on graphene at −800 mV. This particular case is related to a high-quality material with strong near band-to-band emission and lower concentration of defects. As a comparison, the film of ZnO/G/Cu grown at applied potential of −0.9 V shows the lowest I_uv_/I_vis_ value compared to the other films, which suggests that there may be more defects induced during the growth such as oxygen vacancies (V_O_) [57]. In terms of potential applications, Ahn et al. [58] noted that the sensitivity of a gas sensor increases linearly with the concentration of defects in the analyzed samples.

In Figure 13a,b, the Gaussian fitting for the PL spectra of ZnO NWs grown on Cu and ZnO NWs grown on graphene/Cu are presented. Black lines correspond to the experimental data and color lines represent the fitted data. The broad visible emission bands were deconvoluted into three PL peaks. The green emission peak located around 531–535 nm is attributed to the ionized oxygen interstitial and/or complex of V_O_Zn_i_. The green-yellow emission peak around 588–601 nm corresponding to the defects induced by oxygen vacancies (V_O_) situated in the ZnO lattice is the luminescence produced by the recombination of photogenerated holes with electrons of the V_O_ [59]. The yellow-orange emission peak at 635–639 nm is usually attributed to the presence of O interstitials [60,61]. Due to the relative intensities of the three peaks, the center of the visible PL bands emission shows a blue shift for ZnO/G/Cu with respect to the ZnO/Cu, a possible cause for this behavior being the interaction between ZnO nanowires and the graphene.

The emissions spectra of all samples were also characterized using the CIE 1931 chromaticity diagram (Figure 14). The color coordinates of ZnO/Cu were estimated and summarized in Table 3, corresponding to yellow-orange emission. The obtained color coordinates for ZnO/G/Cu display a shift from yellowish-orange to greenish-yellow compared with ZnO/Cu.

## 4. Conclusions

Electrochemical deposition was employed to deposit ZnO nanostructures on copper and graphene/copper working electrodes. The graphene-covered electrodes were obtained by chemical vapor deposition.

Electrochemical deposition of ZnO is a two-step process, i.e., reduction of nitrate ions to nitrite and precipitation of ZnO. The electrochemical measurements show a higher current density in the case of the copper electrodes while the deposition on graphene takes place at more electronegative potentials. The process is one of nucleation/growth, with a higher nucleation rate in the case of copper substrates. The morphological, structural, and optical measurements revealed the fact that the higher structural quality/lower defect concentration can be observed for the samples deposited on graphene, with a higher texture coefficient and intense luminescence given by the exciton recombination.

The sample of ZnO/G/Cu prepared at −0.8 V with the ZnO nanowires diameter of 50–100 nm shows a stronger UV emission at 378 nm as compared to that of ZnO/Cu obtained at the same deposition potential. ZnO/graphene/Cu nanohybrid heterostructures are interesting for a wide range of optoelectronic applications and various sensors.

The use of G/Cu as a substrate is a straightforward preparation procedure, removing the need for a graphene transfer procedure.

## Figures and Tables

**Figure 1 nanomaterials-12-02858-f001:**
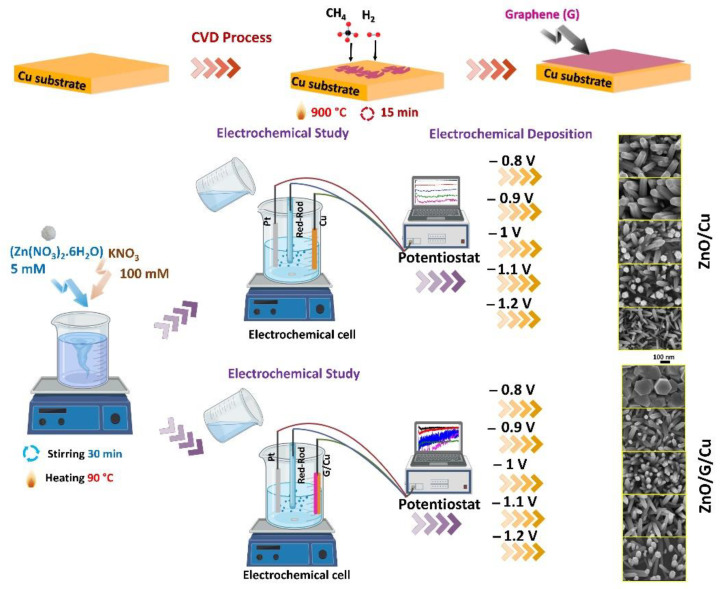
Synthesis process of CVD-graphene, ZnO/Cu, and ZnO/graphene/Cu at different applied potentials.

**Figure 2 nanomaterials-12-02858-f002:**
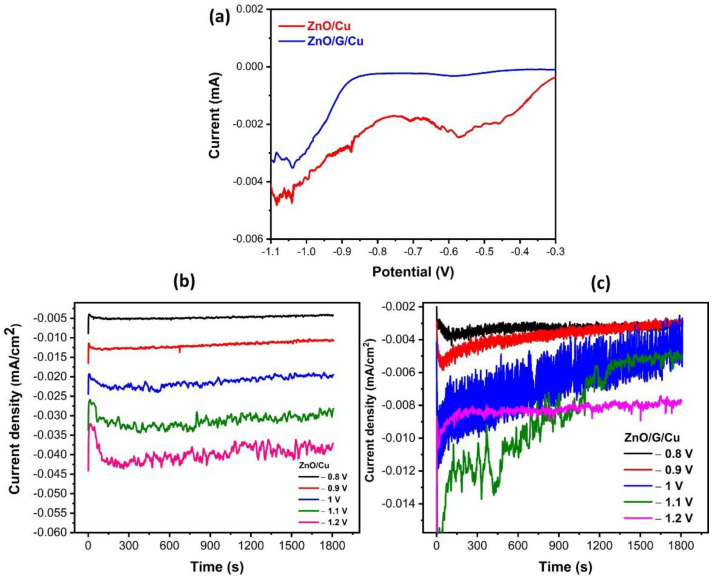
(**a**) Electrochemical polarization curve for ZnO electrodeposition on Cu and G/Cu. Current versus time curves at different deposition potentials for (**b**) ZnO/Cu and (**c**) ZnO/G/Cu.

**Figure 3 nanomaterials-12-02858-f003:**
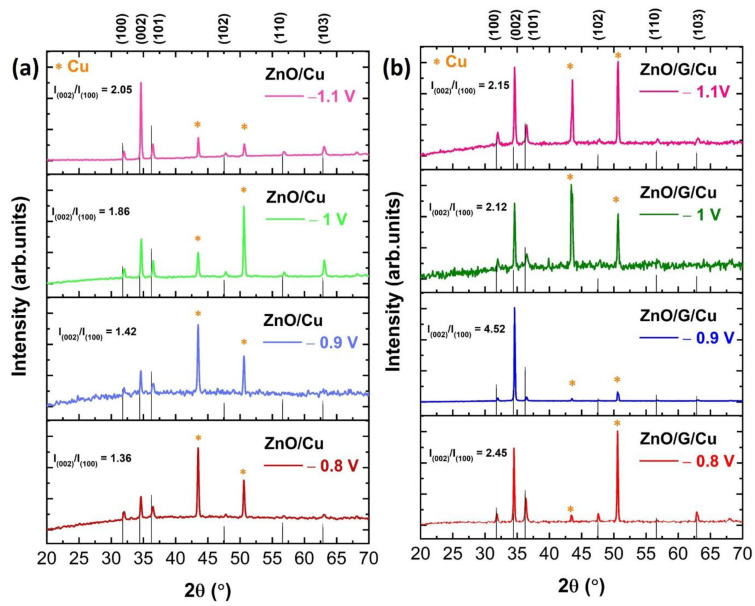
X-ray diffraction patterns of (**a**) ZnO/Cu and (**b**) ZnO/G/Cu electrodeposited at different applied potentials.

**Figure 4 nanomaterials-12-02858-f004:**
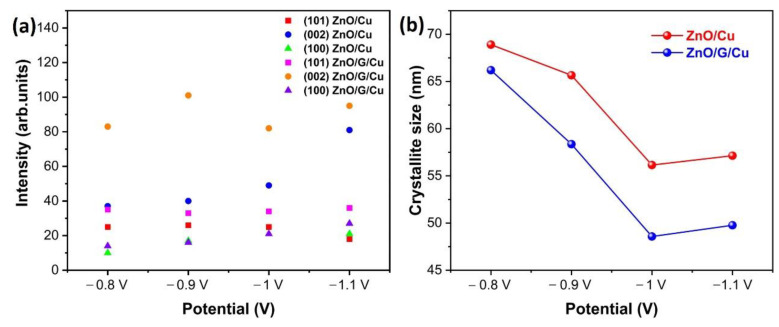
(**a**) The intensity of the (101), (002), and (100) peaks and (**b**) the average crystallite size of ZnO/Cu and ZnO/G/Cu prepared at different applied potentials.

**Figure 5 nanomaterials-12-02858-f005:**
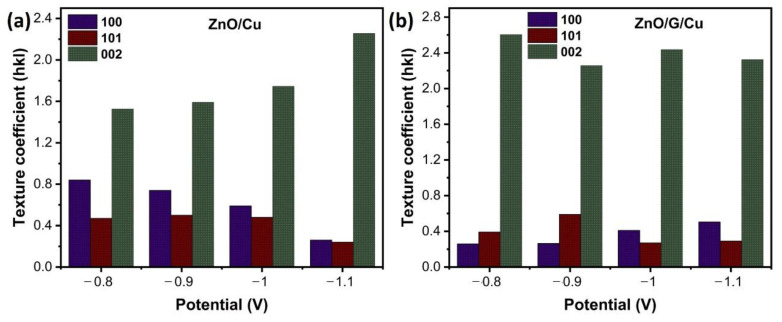
Texture coefficient (TC) of (**a**) ZnO/Cu and (**b**) ZnO/G/Cu at different applied potentials.

**Figure 6 nanomaterials-12-02858-f006:**
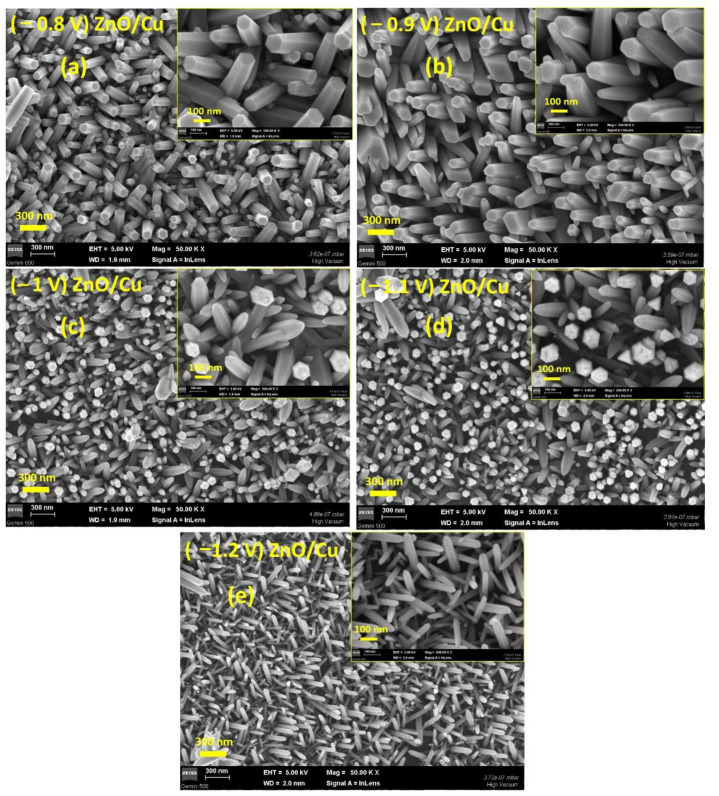
SEM images of ZnO nanowires grown on Cu substrate at different potentials: (**a**) −0.8 V, (**b**) −0.9V, (**c**) −1V, (**d**) −1.1V, (**e**) −1.2 V vs. RE. The insets show the SEM images at high magnification.

**Figure 7 nanomaterials-12-02858-f007:**
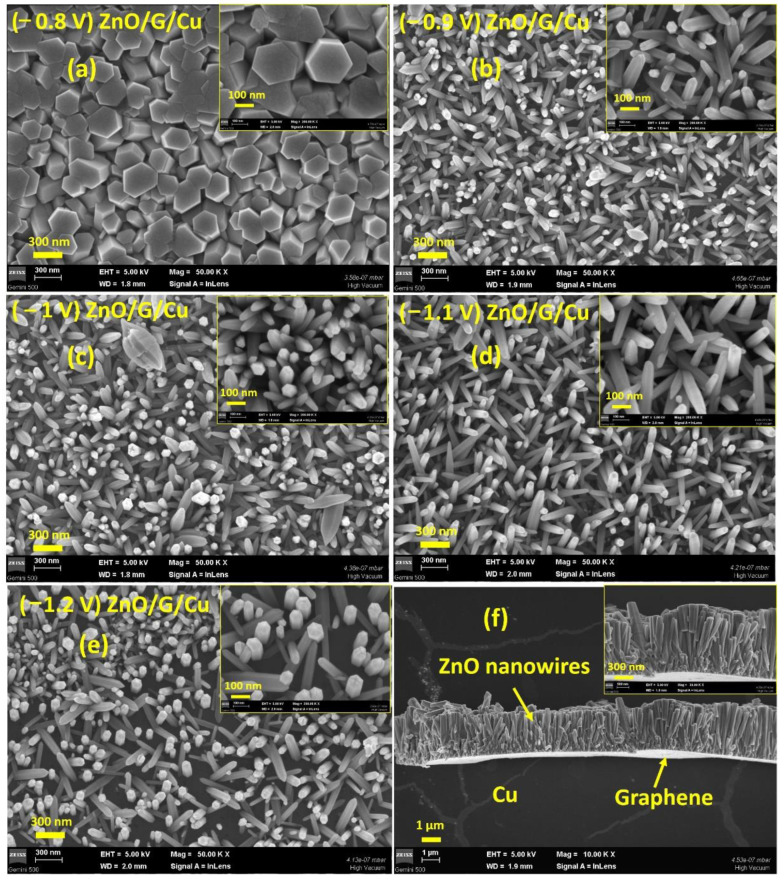
SEM images of ZnO nanowires grown on graphene/Cu at different potentials (**a**) −0.8 V, (**b**) −0.9 V, (**c**) −1 V, (**d**) −1.1 V, (**e**) −1.2 V vs. RE. (**f**) Cross-section of ZnO/G/Cu obtained at −0.9 V vs. RE. The insets show the SEM images at higher magnification.

**Figure 8 nanomaterials-12-02858-f008:**
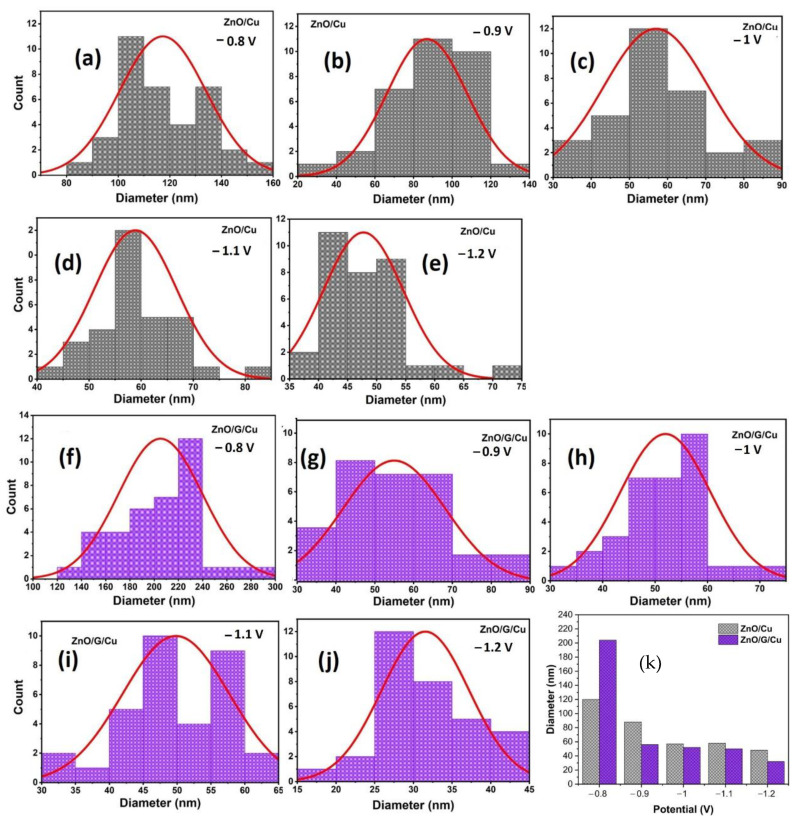
Diameter size distribution of ZnO nanowires deposited on Cu (**a**–**e**) and G/Cu (**f**–**j**) at different potentials, (**k**) average diameter size of ZnO nanowires as function of deposition potential for both types of electrodes.

**Figure 9 nanomaterials-12-02858-f009:**
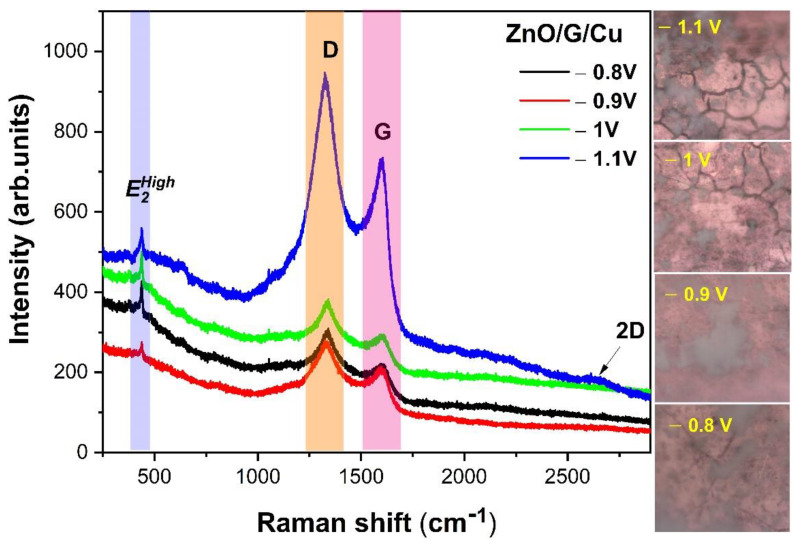
Raman spectra and images of the ZnO/G/Cu films deposited at −0.8 V, −0.9 V, −1 V, and −1.1 V vs. RE.

**Figure 10 nanomaterials-12-02858-f010:**
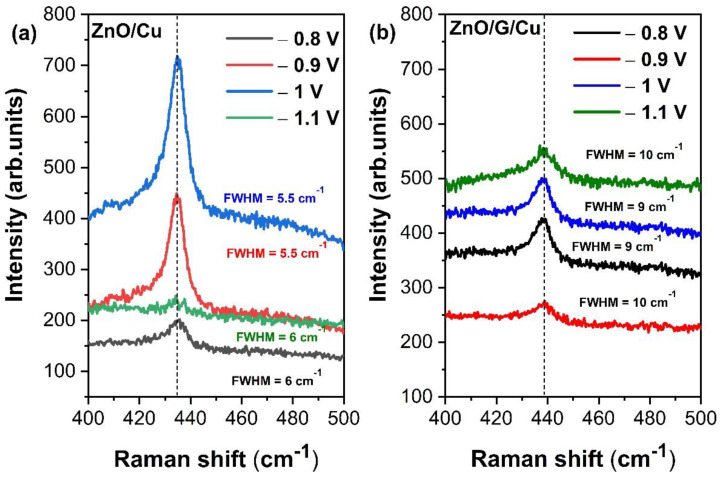
Raman spectra showing enlarged area of $E_{2}^{High}$ (438 cm^−1^) mode positions and FWHM of (**a**) ZnO/Cu and (**b**) ZnO/G/Cu films deposited at different potentials.

**Figure 11 nanomaterials-12-02858-f011:**
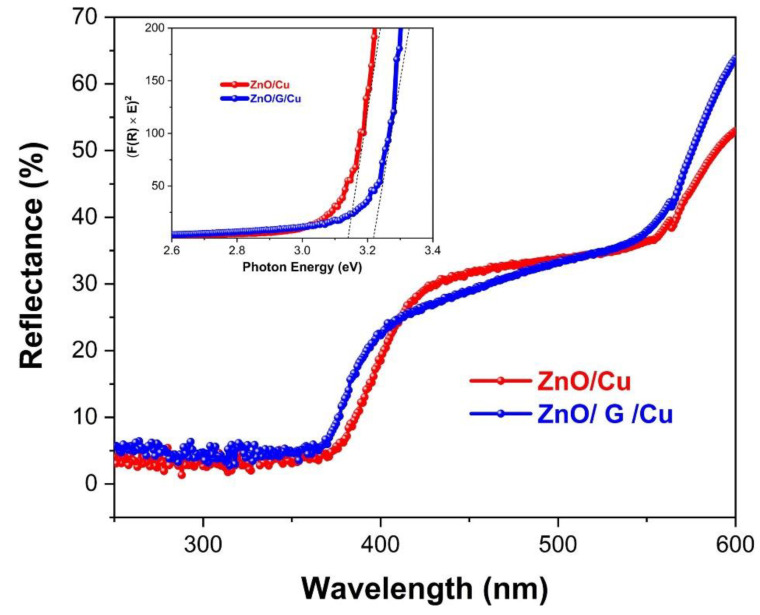
Modified Kubelka–Munk function for direct band gap energy (F(R) × E)^2^ versus photon energy (Inset) and UV–Vis total reflectance spectra of ZnO/Cu and ZnO/G/Cu.

**Figure 12 nanomaterials-12-02858-f012:**
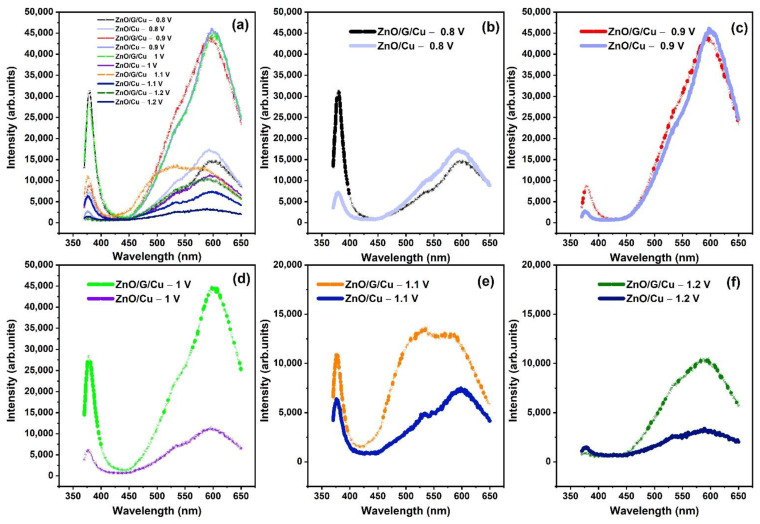
(**a**–**f**) PL spectra of ZnO/Cu and ZnO/G/Cu prepared at different applied potentials.

**Figure 13 nanomaterials-12-02858-f013:**
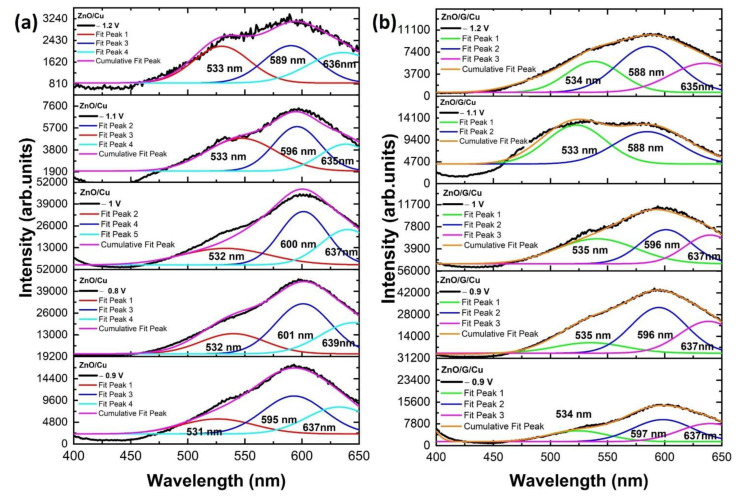
Gaussian fitting of the PL spectra of (**a**) ZnO/Cu, (**b**) ZnO/G/Cu films obtained using different applied potentials (black and color lines represent the experimental and the fitted data, respectively).

**Figure 14 nanomaterials-12-02858-f014:**
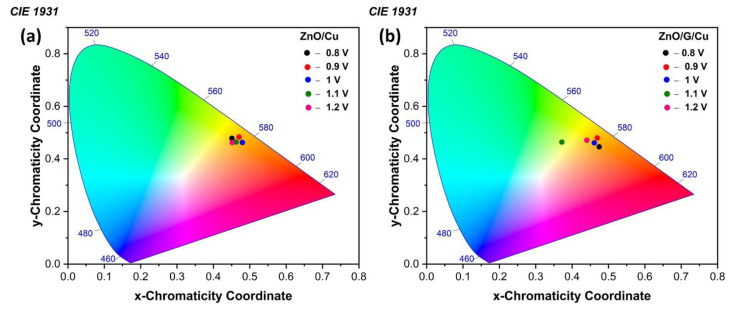
The CIE chromaticity diagram for (**a**) ZnO/Cu, (**b**) ZnO/G/Cu films.

**Table 1 nanomaterials-12-02858-t001:** Values of 2θ, FHWM, crystallite size, and average crystallite size D of ZnO/Cu and ZnO/G/Cu at different applied potentials.

		ZnO/Cu				ZnO/G/Cu			
Potential (V)	(*hkl*) Peak	2θ (Degree)	FHWM (Rad)	Crystallite Size nm	Average Crystallite Size D (nm)	2θ (Degree)	FHWM (Rad)	Crystallite Size nm	Average Crystallite Size D (nm)
−0.8	(100)	31.99	0.156	55.20	68.90	31.98	0.144	63.80	67.03
	(002)	34.6	0.100	86.60		34.57	0.098	85.20	
	(101)	36.45	0.150	64.90		36.48	0.185	52.10	
−0.9	(100)	31.99	0.146	62.30	65.65	31.98	0.232	40.30	58.38
	(002)	34.59	0.112	84.10		34.57	0.091	89.90	
	(101)	36.50	0.163	50.50		36.47	0.220	44.70	
−1	(100)	31.99	0.197	47.50	56.14	31.99	0.264	36.60	48.57
	(002)	34.59	0.129	63.70		34.58	0.122	67.40	
	(101)	36.50	0.173	57.00		36.48	0.240	41.70	
−1.1	(100)	31.99	0.180	53.10	57.13	31.99	0.184	30.10	49.83
	(002)	34.60	0.130	61.20		34.58	0.120	80.20	
	(101)	36.50	0.173	57.00		36.48	0.231	39.10	

**Table 2 nanomaterials-12-02858-t002:** Intensity ratio I_D_/I_G_ and crystallite size (*L_a_*) of G for samples of ZnO/G/Cu.

Potential (V)	I_D_/I_G_	*L_a_* (nm)
−0.8	1.38	27.92
−0.9	1.30	29.64
−1	1.29	29.87
−1.1	1.20	30.03

**Table 3 nanomaterials-12-02858-t003:** Chromaticity coordinates and the PL intensities ratio of the UV emission to the visible emission for ZnO/Cu and ZnO/G/Cu films.

		CIE		I_uv_/I_vis_
Samples	Potential (V)	x	y	
ZnO/Cu	−0.8	0.45	0.47	0.43
	−0.9	0.47	0.48	0.66
	−1	0.48	0.46	0.54
	−1.1	0.46	0.46	0.81
	−1.2	0.45	0.46	-
ZnO/G/Cu	−0.8	0.47	0.44	2.09
	−0.9	0.47	0.48	0.40
	−1	0.46	0.46	0.64
	−1.1	0.37	0.46	0.87
	−1	0.44	0.47	-

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
