# Peer review of "Electrochemical Deposition of ZnO Nanowires on CVD-Graphene/Copper Substrates"

_nanomaterials, 2022, doi:10.3390/nano12162858_

Round 1

Reviewer 1 Report

The authors synthesized ZnO nanowires on copper and graphene/copper with CVD and electrochemical deposition, which costs inexpensively and is easy to control the parameters of materials. They conducted several analyses, such as electrochemical polarization observing the relationship between current density, potential and time for electrochemical study, XRD patterns of the samples for structural properties, SEM images for morphology, Raman spectroscopy for structural quality, reflectance spectra for optical properties and photoluminescence. Most of the plots, images and patterns are simple, understandable and coherent with the discussion in this manuscript; however, some explanations are weird and should be corrected. Also, the overall content looks insufficient in terms of the standards of Nanomaterials. Last but not least, the article is made for sensing application as indicated in the title, but what and how to sense is not clear. Some comments are as follows.

Introduction:

1.        Line 52 typo: based "o"

2.      There is extensive introduction of lots of properties, applications on ZnO, but little on graphene/copper.

3.        It's better not to use the same sentences for lines 27~41 and lines 52~66.

same problem for lines 42~50 and lines 67~76

same problem for lines 81~105 and lines 110~134

Results and Discussion

4.      In Figure 2(b)and 2(c), since it is a comparison, the color of different deposition potentials should be the same in both figures; however, it seems that the pink and the green curves indicate different potentials in Figure 2(b)and(c). Maybe the pink and green curves in figure 2(c) should be swapped because the higher potential sample should have higher current density and more unstable than the other.

5.         At lines 243 to 247, why the result is controlled by reaction rate limitation and diffusion rate limitation? Additionally, please illustrate the mechanisms of reaction rate limitation and diffusion rate limitation. Moreover, why did the crystallite size increase at -1.1V?

6.     Figure 4(b) indicates that the estimated average crystallite size of ZnO/Cu is larger than that of ZnO/G/Cu, while the crystal size of ZnO in figure 7(a) (ZnO/G/Cu) seems to be larger than that of ZnO in figure 6(a) (ZnO/Cu). What’s the parameter used in Scherrer formula here and how come?

7.     At lines 268 to 273, how reaction rate limitation and diffusion rate limitation influenced the morphology of ZnO nanowires should be elaborated.

8.      According to lines 355 to 356, the exciton peak should be more intense at lower electronegative potential but the group of -0.9V shows totally different result where the intensity of exciton peaks are lower than that of group of -1V and -1.1V. Why is that?

9.         What is the reason for showing Iuv/Ivis in Table 2?

10.    Small angle XRD is usually considered more accurate than high angle when it comes to measuring nano scale thin films. Thus, why choosing high angle XRD in this work?

11.    The study can be more rigorous; for example, the experimental data of -1.2V potential is missing in some measurements, such as XRD and Raman.

Author Response

Please find attached our answers to reviewers' comments regarding the paper Electrochemical deposition of ZnO nanowires on CVD-graphene/Copper for sensor application, authors Issam Boukhoubza, Elena Matei, Anouar Jorio, Monica Enculescu and Ionut Enculescu. You will find attached the revised version, where we have highlighted all the changes made in blue color. Also, small corrections were made.  We would like to thank the reviewers for their input, and we hope that as a consequence, the paper reached the standard of quality requested for publication in the journal.

Reviewer 2 Report

Please see the attached file to find the comments.

Author Response

Please find below our answers to Reviewer2’ comments regarding the paper Electrochemical deposition of ZnO nanowires on CVD-graphene/Copper for sensor application, authors Issam Boukhoubza, Elena Matei, Anouar Jorio, Monica Enculescu and Ionut Enculescu. You will find attached the revised version, where we have highlighted all the changes made in blue color. Also, small corrections were made.  We would like to thank the reviewers for their input, and we hope that as a consequence, the paper reached the standard of quality requested for publication in the journal.

Round 2

Reviewer 1 Report

The authors have significantly improved the article.

I recommend the publication of the manuscript.

Reviewer 2 Report

All the reviewer's concern has been addressed.